# Lipid Droplets Are a Physiological Nucleoporin Reservoir

**DOI:** 10.3390/cells10020472

**Published:** 2021-02-22

**Authors:** Sylvain Kumanski, Benjamin T. Viart, Sofia Kossida, María Moriel-Carretero

**Affiliations:** 1Centre de Recherche en Biologie cellulaire de Montpellier (CRBM), Université de Montpellier, Centre National de la Recherche Scientifique, 34293 Montpellier CEDEX 05, France; sylvain.kumanski@crbm.cnrs.fr; 2International ImMunoGeneTics Information System (IMGT®), Institut de Génétique Humaine (IGH), Université de Montpellier, Centre National de la Recherche Scientifique, 34396 Montpellier CEDEX 05, France; benjamin.viart@igh.cnrs.fr (B.T.V.); sofia.kossida@igh.cnrs.fr (S.K.)

**Keywords:** nucleoporins, lipid droplets, genome integrity pathways, fatty acids binding

## Abstract

Lipid Droplets (LD) are dynamic organelles that originate in the Endoplasmic Reticulum and mostly bud off toward the cytoplasm, where they store neutral lipids for energy and protection purposes. LD also have diverse proteins on their surface, many of which are necessary for the their correct homeostasis. However, these organelles also act as reservoirs of proteins that can be made available elsewhere in the cell. In this sense, they act as sinks that titrate key regulators of many cellular processes. Among the specialized factors that reside on cytoplasmic LD are proteins destined for functions in the nucleus, but little is known about them and their impact on nuclear processes. By screening for nuclear proteins in publicly available LD proteomes, we found that they contain a subset of nucleoporins from the Nuclear Pore Complex (NPC). Exploring this, we demonstrate that LD act as a physiological reservoir, for nucleoporins, that impacts the conformation of NPCs and hence their function in nucleo-cytoplasmic transport, chromatin configuration, and genome stability. Furthermore, our in silico modeling predicts a role for LD-released fatty acids in regulating the transit of nucleoporins from LD through the cytoplasm and to nuclear pores.

## 1. Introduction

The nucleus of eukaryotes is circumscribed by the nuclear envelope, which is a specialized subdomain of the Endoplasmic Reticulum (ER) that adopts the form of two double membranes. The two membranes merge at specific regions of extreme curvature, which makes the structure appear to be penetrated by holes. These nuclear pores are scaffolded by large Nuclear Pore Complex (NPC) structures composed of several copies of approximately 30 different proteins, the nucleoporins (Nups). NPCs have a central channel with an eight-fold symmetrical spoke-ring array, which connects to the outward-facing cytoplasmic pore filaments and the inner nuclear basket [1]. NPCs have variable stoichiometry and a partial degree of conservation at the individual protein level, but they have a strikingly conserved structure across species, which is likely reflecting their universal function [2]. This function classically relates to the control of nucleo-cytoplasmic transport, although NPC functions have been expanded in the last decade to the control of gene expression, genome organization, and even DNA replication and repair [3].

Another important structure arising out of the Endoplasmic Reticulum are the Lipid Droplets (LD), which are the only organelles of the cell delimited by a monolayer of phospholipids. LD are classically known for their role as an energy reservoir in the form of, mostly, triacylglycerols (TAG) and steryl esters. However, this oily core varies between cell types and metabolic conditions [4]. Present in virtually all species [5], LD form around neutral lipids that coalesce within the two leaflets of the ER membrane, frequently the cytoplasmic one. Once a threshold concentration is reached, which is governed by the membrane lipid and protein composition and the type of neutral lipids themselves, the lipids condense in a spherical form and bud out into the cytoplasm [6,7,8]. LD emancipation from the ER can either be complete, or partial, if they remain anchored by their base to the ER’s external leaflet [9,10]. Furthermore, enrichment within the membrane of specific lipids, especially those favoring negative curvature, may embed LD within the two ER leaflets [11,12,13,14]. As well as functioning as an energy reservoir, LD shelter lipids from unscheduled oxidation and also protect the cell from lipotoxicity [15,16]. Moreover, the LD surface is covered by a rich proteome, which is under constant evolution [17,18,19]. A core resident proteome maintains the function of the LD, but there is also a selective reservoir of other proteins residing on them. Storing proteins on LD allows them to be made available elsewhere in the cell, in a regulated manner [20], for example as assembly platforms for viruses or to store antiviral proteins [21,22], or even to assist the dissolution of protein aggregates [23].

LD contain several proteins fulfilling important roles in nuclear biology. The most prominent and conserved example is histones. Histones are maternally deposited onto Drosophila embryo LD, and their timely release sustains correct cell division. Indeed, histone overexpression accelerates nuclear division, while their depletion forces their unscheduled expression by the embryo [24,25]. Histones have also been found in LD proteomes from other species, although their physiological role remains unassessed [26,27]. In mouse, the nuclear DNA repair and splicing factor Prp19 was found on adipocyte LD, where it appears to have an additional role in LD biogenesis [28]. Furthermore, LD can harbor and dictate the availability of transcription factors, such as the hypertonic stress-responsive Nuclear Factor of Activated T-cells 5 protein [29] or the MAX Dimerization Protein MLX [30]. Together, these observations raise the notion that the storage of nuclear factors on LD may fine-tune their availability in the nucleus. However, few studies have assessed this possibility, and LD proteome analyses are rarely oriented into this direction. In this work, we exploited publicly available LD proteomes to actively search proteins important for nuclear homeostasis. Importantly, we demonstrate that a subset of Nups resides on LD. In addition, we provide evidence that genetic defects that alter the timely deposition of Nups on LD sensitize cells to additional defects in nuclear pathways. From a physical perspective, if LD formation is inhibited, Nups overpopulate the nuclear membrane with NPCs, and this expands it. Finally, we present in silico predictions and genetic evidence suggesting that fatty acids binding is a means to modulate Nup transit between LD, the cytoplasm, and the NPC.

## 2. Materials and Methods

**Reagents** used in this work were BODIPY (790389, Sigma-Aldrich, Saint-Quintin Fallavier, France), oleic acid (O1008, Sigma-Aldrich, Saint-Quintin Fallavier, France), inositol (PHR1351-1G, Sigma-Aldrich, Saint-Quintin Fallavier, France), cerulenin (SC-200827A, Santa Cruz Biotechnology, Dallas, TX, USA), and AUTODOT^TM^ (SM1000a, Abcepta, San Diego, CA, USA).

***S. cerevisiae* cell culture and treatments**: Cells were grown at 25 °C in YEP (rich) or yeast nitrogen base (YNB) (minimal) medium supplemented with 2% glucose (dextrose), unless indicated otherwise. Transformed cells were selected for plasmid maintenance in YNB–leucine or YNB–uracil medium. To induce the overexpression of *DGK1* and *PAH1-7A*, cells were grown overnight in YNB–leucine with 2% glycerol. The next morning, 2% galactose was added for 3 h prior to imaging cells for the experiment. For the oleate experiments, a pre-culture was set early in the morning of day 1. On the one hand, this pre-culture was used in the evening to inoculate new cultures either in the presence 0.05% oleic acid (condition “Oleate”) or nothing (condition “exponential”), allowing them to grow as to have exponential cultures (5–7 × 10^6^ cells/mL) on the morning of day 2. Alternatively, the pre-culture was left to grow, yielding a saturated culture on the morning of day 2 (3 × 10^8^ cells/mL), which was used for the condition “saturated”. For the cerulenin experiments, saturated cultures (3 × 10^8^ cells/mL) were mildly diluted (1.5 × 10^7^ cells/mL) with fresh medium in the presence (10 μg/mL) or not of cerulenin, and pictures were taken at the indicated times. For the inositol experiments, cultures were set on the morning of day 1 in the presence (75 μM) or not of inositol and left to grow until saturation for at least 24 h.

**Fluorescence Microscopy:** First, 1 mL of the culture of interest was centrifuged; then, the supernatant was thrown away and the pellet was resuspended in the remaining 50 μL. Then, 3 μL of this cell suspension was directly mounted on a coverslip for immediate imaging of the pertinent fluorophore-tagged protein signals. To dye LD, 1 mL of a 100 μg/mL BODIPY or 1 μL of a 1 mM stock AUTODOT^TM^ was added and mixed to the 50 μL of centrifuged pellet (with residual medium) prior to mounting. Imaging was achieved using a Zeiss Axioimager Z2 microscope and visualization, co-localization, and analysis performed with Image J v2.0.0-rc-69/1.52i.

**LD data curation and analysis:** The hits and the spectral counts reported in Figures 1A, 6A, and 7A were retrieved from [31]. We manually curated all the reported hits from wild-type (WT) and mutant LD proteomes and selected only those relating to nuclear biology. We further classified them into functional categories after individual verification of the best-defined protein function (Appendix A).

**Genetic interactions curation:** All the information regarding the genetic interactors was retrieved from the *Saccharomyces* Genome Database. The information regarding the mutants sensitive to oleate was retrieved from [32]. The selection of nuclear interactors among the totality of annotated hits was performed manually with the criterion that the concerned protein has an activity well-known to impact nuclear homeostasis. We excluded hits involved in mitosis and specific transcription factors whose target(s) are a small gene subset. The gene sets were organized into functionally related thematic groups of interest using the DAVID website [33]. First, we visualized the yielded groups as heatmap plots that were produced using R. Upon validation of their coherence, the clouds presented in the figure were drawn manually for easier visualization.

**Graphical representations** were made with GraphPad Prism to both plot graphs and statistically analyze the data. For data representation, the SEM (standard error of the mean) was used. The SEM estimates how far the calculated mean is from the real mean of the sample population, while the SD (standard deviation) informs about the dispersion (variability) of the individual values constituting the population from which the mean was drawn. Since all the measurements we were considering for each individual experiment concerned a mean (the percentage of cells in the population bearing Nup puncta), and the goal of our error bars was to describe the uncertainty of the true population mean being represented by the sample mean, we did the choice of plotting the SEM.

**Image quantification** to determine the Nup49-GFP (Green Fluorescent Protein)-defined nuclear area presented in Figure 5B was performed using ImageJ v2.0.0-rc-69/1.52i. Briefly, a threshold was manually established for each image using a convenient ImageJ algorithm, making sure that the real signals were the most accurately possibly defined. Thus, “Analyze Particles” was run to retrieve the area of each selected object. Image analysis, to determine the number of NPC clusters observed in each central plane of Figure 5C, was performed manually upon visual inspection of images and following the guidelines presented by [34,35]. Last, the percentage of cells displaying fragmented nucleoli, as identified by far-from-the-real-nucleolus Nop1-CFP (Cyan Fluorescent Protein) signals, was established by visual inspection following the criteria described by [36].

**Fatty acid-binding pocket prediction** was performed with PickPocket, as described [37]. We define cavity as the structural “hole” in a protein, while pocket is a computational construct used to analyze protein cavities. As such, the pocket size and shape depend on the parameter and the software being used. In our case, two predictive models were trained using two manually curated sets of structures containing different fatty acids (FA) or structurally similar ligands. One set contains 340 structures of human proteins only (Human Model See FA_MannuallyCurated_Train.tsv), and the second set contains 70 structures from 52 different species (All Species Model FA_AllSpecie_Train.tsv), which are both available as Appendix A. During this training, we used a table of real FA-binding pockets, associating them to the value of 1, and a table of non-binding pockets, associating them to the score 0. For each structure, all pockets and secondary structures were computed using fpocket (with default parameters except for −m 4 and −D 6, so the pockets better fit the FA cavities) [38] and Stride [39], respectively. Then, we trained a neural network multilayer perceptron [40] to predict whether a cavity is a “true” pocket (i.e., can interact with ligands of the family under study) or a “false” pocket. To avoid overfitting, both models were trained using a 5-fold cross-validation and reached accuracy above 90%. We used the 3D structures of the concerned Nups from the Protein Data Bank [41] and computed the predictions using the two models; for each pocket of those proteins, a score between 0 (negative match) and 1 (positive match) is provided. We arbitrarily set the cutoff at 0.5 and discarded any hit receiving a lower score. To measure the false positive rate (FPR) of the model, we used the BioLiP database [42] and removed the ligands that were part of the trained model (File “LigandTableFA.xls available as Appendix A). There were 43,392 unique Protein Data Bank (PDB) IDs in this set, each having at least one ligand in a cavity. We took a random sample of 1000 PDBs (generated using the shuffle function as implemented in the GNU/linux kernel of Ubuntu 20.04, available at https://man7.org/linux/man-pages/man1/shuf.1.html (accessed on 1 January 2021)) from this set and computed a total of 27,222 pockets in them. Running the “All Species Model”, which is less stringent, threw a total of 11,643 positive (score > 0.5) pockets, i.e., with an FPR of 42%. Running the prediction using the “Human model” yielded 1281 positive pockets, meaning an FPR of 4.7%. Visual inspection of the actual cavities revealed that ligands possessing an aromatic or cyclic structure have a propensity to be identified as positive. Importantly, the common FPR upon crossing both predictions was of 3.1%. Therefore, for the sake of coherence, we only considered for analysis in this work the hits commonly predicted by both models. Images were generated with PyMol (Schrödinger Inc., New York, NY, USA) and the corresponding .pse files (4HMC_Nup157.pse and 2QX5_Nic96.pse) are available as Appendix A.

**Strains and plasmids:** To build a Nup57-tDimer strain tagged at its genomic locus, the plasmid pRS305-Nup57-tDimer [35], kindly provided by O. Gadal, Toulouse, was linearized with *Bgl*II and inserted by homologous recombination in an otherwise WT strain. To build the plasmid expressing the fragment corresponding to the Nic96 fatty acid-binding pocket (FAB) upstream of mCherry, which is a pRS316 vector already bearing the promoter of *CYC1*, the gene coding for mCherry, and the terminator of *NUP1,* was linearized by restriction in between the promoter of *CYC1* and the gene coding for mCherry. The fragment harboring Nic96^FAB^ (aminoacids 281 to 434 from Nic96) was amplified from genomic DNA using the following primers: **forward:** TACTATACTTCTATAGACACACAAACACAAATACACACACTAAATTAATAtctagaatgcagtttttacaatataccg; **reverse:** CTCATAAATTCTTTAATAATAGCCATATTATCTTCTTCACCTTTTGAAACGGATCCttcaatgcttaaagtaactgc, which are homologous at their 5′ ends with the 3′ extremity of the *CYC1* promoter and the 5′ end of the mCherry gene, respectively. The linearized vector and the PCR product were transformed in the Nic96-GFP strain, and the colonies undergoing plasmid reconstitution through homologous recombination (in vivo cloning) were selected on -ura plates. The strains used in this study are presented in (or were derived by crossing/transformation of those presented in) Table 1. Mutants were obtained either by classical gene disruption or crosses. The plasmids used in this study are presented in Table 2.

## 3. Results

### 3.1. Nucleoporins Localize to LD in Saccharomyces cerevisiae Cells

In *S. cerevisiae*, LD-resident proteins have been defined by stringent purification methods [26,43,44,45]. To gain new knowledge about the functional yet transient association of proteins with LD, we studied publicly available datasets where purification stringency was decreased while maintaining high isolation quality standards. We manually curated and classified, into functional modules, the hits retrieved from cytosolic LD purified from WT cells in stationary phase [31] and focused exclusively on hits relating to nuclear biology (Appendix A). We noticed the outstanding presence of a subset of nucleoporins (Nups) at levels comparable to those of the bona fide resident LD lipases, Tgl3 and Tgl5 (Figure 1A). In particular, the enriched hits belonged to the inner pore complex, including the transmembrane anchor Pom152, and almost all the inner adaptor complex Nup170–Nup157, plus some central phenylalanine-glycine-(FG)-containing Nups including Nsp1, Nup49, and Nup57. Contrarily, cytoplasmic or nucleoplasmic basket proteins were poorly represented or even absent (Figure 1A).

In *S. cerevisiae*, LD are created from and remain linked to ER membranes [46]. A direct explanation for the LD proteome Nup data would be that inner pore complex Nups, the most closely associated to the ER, are simply contaminants from LD purification. To test this possibility, we monitored the localization of several Nups belonging to different parts of the NPC by fluorescence microscopy, including Nup188 and Nic96 from the inner pore channel, two of the FG central ring components, Nup49 and Nup57, whose presences in the proteome dataset were modest and the cytoplasmic Nup159. Wild-type (WT) cells grown to saturation overnight in rich medium indeed displayed cytoplasmic, foci-like signals that clearly differed from the nuclear rim-associated ones. This was true for all tested Nups, including the cytosolic Nup159, which was not retrieved in the LD proteome purification (Figure 1B). Between 20% and 30% of the cells had these cytosolic puncta, which clearly co-localized with LD, as observed both in Differential Interference Contrast (DIC) images and upon LD staining with the vital dyes AUTODOT^TM^ or BODIPY (Figure 1B). Of note, not all AUTODOT^TM^ or BODIPY-positive signals were Nup-positive, ruling out a channel bleed-through phenomenon. Last, we noticed that LD-coincident Nup signals were frequently cortical, far from the nuclear ring (Figure 1B). Altogether, these data suggest that at least a subset of Nups reside on LD during the stationary phase.

### 3.2. LD Are a General Reservoir of Nucleoporins

Nups in cytosolic foci in *S. cerevisiae* have already been reported in the literature [47,48,49,50,51]. While in one of those mentions, the authors described the intimate association of these structures with the Endoplasmic Reticulum, the authors did not draw the inference that these structures were indeed LD [49]. Of note, these cytoplasmic Nup foci were only observed in mutant contexts, suggesting that the association of Nups with LD may be dynamic and respond to specific cues. To investigate this, we quantified the percentage of WT cells with Nup cytoplasmic foci when growing exponentially in rich yeast peptone dextrose (YPD) medium. The percentage of positive cells in the population significantly dropped to 2–12% during the first hours after cells had been diluted, when they resumed cycling. However, Nup cytoplasmic foci increased, to being present in 50% of the cells, in the late exponential phase (Figure 1B, quants). This behavior is fully reminiscent of LD lipolysis and re-formation kinetics [52]. Since cell cycle resumption is accompanied by lysis of LD contents and their concomitant shrinkage is known to evict its residing proteome [53], it is likely that LD may act as depots for the concerned Nups under no-cycling conditions.

Therefore, we hypothesized that forcing the cellular consumption of LD contents by adding cerulenin [46,52] should accelerate the kinetics of disappearance of Nup cytosolic puncta. We added cerulenin at mild culture dilution; 3 × 10^8^ cells/mL cultures were diluted to 1.5 × 10^7^ cells/mL (as described in [54]). As anticipated, cerulenin reduced the number of Nup cytoplasmic puncta (Figure 2A). Furthermore, from 2h treatment and onwards, progressively, more and more cells displayed diffuse fluorescent signals in the cytoplasm for all tested Nups (Figure 2B). In extreme cases, such as that of Nup159, this cytoplasmic signal correlated with disappearance of NPC-associated signals (Figure 2B). These data suggest the following: (1) the NPC may progressively absorb Nups that are released from LD, with a physiological timing during re-entry into cycling conditions; (2) the sudden consumption of LD, induced by cerulenin, may force an unscheduled release of Nups into the cytoplasm, suggesting they cannot be accommodated at the NPC; and (3) fatty acids may stabilize Nups close to membranes. Together, our data suggest that LD serve as a physiological reservoir for Nups during non-cycling conditions.

### 3.3. Mis-Regulation of LD Biology Impacts Nup Physiology

LD formation depends on the nutritional and growth status of the cell. Molecularly, the abundance and the type of phospholipids in the Endoplasmic Reticulum bilayer determine the ability of LD to bud toward the cytoplasm and become independent structural entities [7,14]. To further explore how LD-forming drivers condition Nup storage, we returned to the LD proteome data [31], but this time, we analyzed *S. cerevisiae* cells with impaired LD formation. For example, we analyzed *fld1Δ* cells, lacking seipin, the underlying cause of Berardinelli–Seip congenital syndrome, which is the most severe case of lipodystrophy in humans. Seipin functions to channel and coordinate the flow of neutral lipids into LD. Seipin knockouts have entangled and unproductive LD due to lack of a continuous neutral lipid flow. Additionally, the sudden confluence of locally increased unpacked neutral lipids eventually gather to generate a massive LD that is 30 times bigger than normal [54,55,56]. Another protein we analyzed was Cds1. Cells lacking Cds1 are deficient in phospholipid synthesis. As a consequence, neutral lipids are stored in oversize LD, which from a physical perspective will permit saving membrane by increasing the volume-to-surface area ratio. In our proteome analysis, we observed that every concerned Nup decreased on LD, in some cases to undetectable levels, in both mutants (Figure 1A).

When assessing this matter experimentally by monitoring Nup49 and Nup57, we found first that, contrary to the proteomic data, cells deficient in phospholipid synthesis (*cds1Δ*) displayed slightly more Nup cytoplasmic puncta coincident with LD than WT cells (Figure 3). In the absence of Cds1, the lower phospholipid density of the LD monolayer provokes packing defects, whereby neutral core contents are partially exposed to the cytoplasmic environment. Thus, we speculated that this unexpected Nup binding may arise from a thermodynamically favorable reaction that compensates for these packing defects [6]. In support of this notion, growing WT cells exponentially in the presence of 0.05% oleate, to artificially induce LD packing defects [44], increased the detection of LD-associated signals for Nup49-GFP and Nup57-tDimer at levels comparable to those observed in stationary cultures (Figure 4A).

Regarding *fld1Δ* cells, we found similar results to those derived from proteomic analyses, namely less cells in the population displaying Nup cytoplasmic puncta coincident with LD (Figure 3A). We attribute the small and entangled LD phenotype of *fld1Δ* cells to the role of seipin in initiating and channeling neutral lipids for storage, while the giant LD phenotype is due to the stochastic blebbing of the accumulated neutral lipids out of the ER, once a threshold concentration has been reached. The giant, but not the entangled, LD phenotype can be suppressed by inositol [54,55], offering us the opportunity to dissect the cause of Nup under-representation on LD in seipin-deficient cells. If low Nup levels were due to super-sized LD restricting Nup loading, inositol addition would be expected to restore cytoplasmic puncta back to (or close to) WT levels. Instead, if the low Nup level was due to a lack of proper LD formation, inositol supplementation would fail to restore Nup cytoplasmic puncta. The addition of inositol effectively suppressed the giant LD phenotype of *fld1Δ* cells yet failed to increase the low percentage of cells in the population with Nups on LD (Figure 3). Inositol supplementation slightly but recurrently decreased the percentage of cells harboring Nup on LD in all cells, including *cds1Δ* cells and WT (Figure 3B). Importantly, in *fld1Δ* cells, where the entangling of small LD is increased by inositol supplementation [55], the deformation of the NPC-defined nuclear rim was very strong (Figure 3B, yellow frame).

These data argue that the impaired Nup deposition onto LD mostly stems from any impediments LD may encounter to protruding from the ER. Another means to entangle LD, through nuclear membrane deformation, is by locally elevating phosphatidic acid by overexpression of the diacylglycerol kinase Dgk1 [56]. Two observations supported the notion that Nup deposition on LD is mainly conditioned by the LD’s ability to emerge from the ER. Firstly, we observed that *DGK1* overexpression, which greatly deformed the nuclear rim (both monitored by the ER transmembrane marker Sec63 (Appendix A) and the Nup-defined rim itself (Figure 4B)), mildly but repetitively decreased Nup presence on LD. The second confirmation came when we reciprocally drove phosphatidic acid consumption by overexpressing the hyperactive *PAH1-7A* allele of lipin, which boosts LD biogenesis [57]. Driving the consumption of phosphatidic acid slightly increased the frequency of Nups on LD (Figure 4B), and it dramatically increased the number of cytoplasmic puncta per cell (Figure 4C). Altogether, we conclude that Nup deposition on LD mainly relies on the ability of LD to correctly pack within and escape from the ER.

### 3.4. NPCs Overload at the Nuclear Membrane When LD cannot Form

The correlation between LD formation and Nup deposition on LD prompted us to wonder what happens to Nups in cells that cannot generate LD. We used a strain that cannot esterify TriAcylGlycerols (TAG) or generate steryl esters (*dgaΔ lro1Δ are1Δ are2Δ*, from now on named *4Δ*), in which Nup49 was GFP-tagged. Whether grown to saturation or during exponential growth, these cells rarely displayed any Nup49-GFP-associated signals in the cytoplasm, except in less than 1% of the cells. In these few cases, the signals, which of course did not co-localize with any LD, were less rounded and more fragmented in aspect (Figure 5A). In the remaining 99%, Nup49-defined nuclear rims had a “healthy” round appearance in the *4Δ* mutant, which was indistinguishable from the WT (Figure 5B, left). Nevertheless, we noticed that the area defined by the Nup49-GFP signals (the nucleus) was frequently enlarged (Figure 5B, right). Furthermore, we inspected Nup49-GFP clusters at the nuclear rim. Clusters are known to originate from the non-random, focal distribution of three to nine nearby NPCs within the nuclear envelope [34,58]. Consistent with these past reports, WT cells averaged nine clusters in the central nuclear plane (Figure 5C). Importantly, the *4Δ* mutant recurrently showed a higher density, with 11 to 13 NPC clusters (Figure 5C). These data support the notion that the scarcity of LD increases Nup presence at the nuclear envelope. This excess likely expands the nuclear envelope and packs it more densely with NPCs. Overall, then, we conclude that LD formation permits the storage of Nups on LD, which act as buffer, and that any process harming LD formation puts this Nup depot at risk.

### 3.5. Nucleoporin Redistribution onto LD Reconfigures Nucleo-Cytoplasmic Transport

To explore the potential impact of Nup redistribution on cell physiology, we looked for other changes in LD proteomes of *cds1Δ* and *fld1Δ* cells. We noticed that classic Nup interactors and essential players in nucleo-cytoplasmic transport, namely karyopherins, whose presence was negligible on WT LD, appeared as markedly present on the mutant cells LD (Figure 6A). We used fluorescent Dendra2-tagged karyopherin Kap123 in WT and mutant strains to further study this. In a WT strain growing exponentially, Kap123 signals were diffusely cytoplasmic and nuclear, with a more marked nuclear intensity in rich medium (Appendix A). This behaviour was also seen in *fld1Δ* cells. In contrast, exponentially growing *cds1Δ* cells had only faintly vacuolar Kap123 signals or no signal at all (Appendix A). Instead, in saturated cultures of cells grown in minimal, defined medium, Kap123–Dendra2 signals had similar cytoplasmic diffuse and puncta-like signal patterns in all three strains (Appendix A). As a result of this common pattern, and because minimal medium exacerbates the phenotypes associated with the lack of Cds1 and Fld1 [54], we next used this medium. Kap123 cytoplasmic puncta accumulated in approximately 70% of cells in all three strains (Figure 6B). Surprisingly, these punctate structures did not co-localize with LD in WT cells (Figure 6C, pink arrowheads). It remains to be investigated which sub-cellular structures these sites represent. Importantly, however, and in agreement with the proteomic data, Kap123 signals partly (*cds1Δ*) or completely (*fld1Δ*) co-localized with LD in the mutants (Figure 6C, yellow arrowheads).

Karyopherins play a role in the nuclear import of proteins related to RNA metabolism, chromatin, and nucleolar biology. In line with these functions, we found a strong enrichment of karyopherin cargoes belonging to these categories exclusively on the proteomes retrieved from mutant LD (Figure 7A). These results suggest that LD contribute to the correct dosage/distribution of Nups and their interacting partners, the karyopherins, thus impacting nucleo-cytoplasmic transport and other NPC-controlled processes. If this were the case, we would expect *cds1Δ* and *fld1Δ* mutations to interact genetically with mutations in mRNA, rRNA, sn(o)RNA, and chromatin networks. The data in Figure 7A suggest just this could be the case. Using available data in the *Saccharomyces* Genome Database, we classified the functional modules to which the nuclear genetic interactors of *cds1* and *fld1* cells belong. The nuclear genetic interactions of these mutations indeed comprise all RNA biology aspects from transcription to processing, rRNA and ribosome assembly, as well as the chromatin landscape (Figure 7B, left and center). Reciprocally, since exponential growth in oleate seems to drive the inappropriate deposition of Nups onto LD (Figure 4A), we scored whether transcription, nucleolar ribosome assembly, and chromatin mutants are sensitive to oleate, as reported in [32], which was found indeed to be the case (Figure 7B, right).

To assess this experimentally, we asked whether *cds1Δ* and *fld1Δ* cells are defective in one of the enriched pathways, namely nucleolar biology, as deduced from the enrichment in the hits “snoRNA”, “rRNA”, and “chromatin”. A fast, general, and informative means of assessing nucleolar stability is to monitor the integrity of Nop1 signals. Nop1 is the yeast ortholog of fibrillarin, which is an exclusively nucleolar histone H2A glutamine methylase [60] that, when tagged and monitored by fluorescence live microscopy, provides a readout for nucleoli status. For example, active rDNA transcription during exponential growth in rich medium leads to nucleolar deployment, while low rDNA transcription during starvation leads to compact nucleoli [61]. In the case of nucleolar instability, which can arise from a variety of defects, Nop1 signals can give rise to different aberrant patterns, e.g., nucleolar fragmentation in the shape of foci distant from the bona fide nucleolus [36], nucleolus splitting into two unconnected bodies, each harboring one rDNA sister chromatid [62], nucleolar enlargement because of deficient chromatin compaction [63], and detachment from the inner nuclear membrane and migration toward the interior of the nucleus [64]. Importantly, we found that WT cells displayed a negligible level of nucleolar instability (around 1% of cells), while *cds1Δ* and *fld1Δ* cells had a 7-fold and 10-fold increase in nucleolar instability, respectively, which was manifested in the shape of distant-from-the-nucleolus Nop1 foci (Figure 7C).

Clearly then, genetically or pharmacologically affecting LD formation impacts the biology of chromatin, nucleolar metabolism, and transcription, and this likely sensitizes cells to additional mutations affecting nuclear pathways. Given the results from the previous sections, we suggest that LD affects the nucleus specifically through the modulation of NPC properties.

### 3.6. Nucleoporins Detachment from LD may Be Coupled to the Release of Fatty Acids

Nups associate with LD through different features. Nuclear pore complex proteins such as the transmembrane protein Pom152 possess a single transmembrane pass and a very long, globular tail protruding in the ER lumen [65]. This may help it concentrate at the base of LD, which remain attached to the ER, as recently proposed [66]. Soluble Nups, on the other hand, may anchor to the LD monolayer by deploying amphipathic helices, as described for several of them [67,68,69,70]. To unveil additional features that could dictate whether Nups adhere to LD or not, we built on recent data showing that, during lipolysis, lipase-released fatty acids (FAs) bind LD-coating proteins, such as Perilipin 5, thus permitting its translocation into the nucleus [71]. Furthermore, we were prompted by our own data from Figure 2, indicating that FA release by cerulenin induces a dislocation of Nups into solubilized cytoplasmic forms. Thus, we took advantage of our recently developed PickPocket tool [37] to ask whether any of the Nups on wild-type LD for which the structures are available (Figure 8A, fourth column) have potential FA-binding pockets.

Based on a model built both with positive FA-binding pockets and unbound (negative) pockets from multiple species (Appendix A), PickPocket predicted pockets in three different Nups (Nic96, Nup157, and Nup192), and three potential pockets at the interface of Nup116 assembled with its partners (Figure 8A). To increase stringency, we ran PickPocket again using a model exclusively trained with positive and negative pockets of human origin (Appendix A). With the exception of Nup192, the same hits were retrieved (Figure 8A), and the predicted pockets’ location was identical between both models regarding Nic96 and Nup157 (Figure 8A). Both of these proteins are central anchoring subunits structuring the NPC from its most inner part. Therefore, we concentrated exclusively on these two robust hits. Nup157 is highly flexible at defined hinges where the molecule bends to adopt alternative conformations. This facilitates its role as an adaptor accommodating transport through the pore channel [72]. In particular, a C-shaped_70-893_ domain, where a β-propeller segment interacts with a α-helical fragment, is maintained by a well-defined hydrophobic interface stabilized by the pairings between Phe97–Phe709, Leu101–Ala713, Phe590–Tyr635, and Tyr594–Tyr646 [72]. The FA-binding pocket we predict (residues between 594–762) fully coincides with this interface (Figure 8B, filled structure). In particular, the apolar tail of the FA would directly sit on the contact established between Tyr594 and Tyr646 (Figure 8B, magenta and orange residues, respectively), and it therefore has the potential to impact the C-shape of the protein by modulating this hydrophobic interaction (Figure 8B, inset).

Nic96 is the most abundant protein of the NPC. Nic96 is essential for pore assembly, as pores are defective in its absence, the size exclusion barrier of transported cargoes is decreased, and transcriptional patterns are altered [58,73,74]. Nic96 is elongated, with a curved N-terminus, and distinct head, central, and tail modules [50,75]. We define the patch comprising the pocket (281–434) as fully coinciding with Nic96 head (268–478) (Figure 8C). In particular, the potential FA-contacting residues we map, Q281, F282, and Y285 in Helix 4; F308 and K312 in Helix 5; L430, S431, E433, and D434 in Helix 11, are all highly conserved between species and qualified as structurally important [75]. Both the N-terminus and the head of Nic96 appear essential for its localization at the NPC for different reasons. The N-terminus ensures the interaction of Nic96 with its partners and thus its stabilization at the NPC. However, the head domain, in whose absence Nic96 still interacts with its partners, prevents Nic96 from staying blocked at cytoplasmic spots that very likely represent LD [50]. Together with our data showing that cerulenin triggers the solubilization of Nup signals (Figure 2) and our prediction that Nic96 has a potential FA-binding pocket in its head domain (Figure 8), we were prompted to postulate that the presumed FA-binding activity of Nic96 may serve to attach it to lipolysis-released FAs, thus making this Nup available to reach the NPC.

To test this possibility genetically, we expressed either mCherry alone or Nic96^FAB^ (for Nic96 Fatty Acid Binder)-mCherry in cells in which full-length Nic96 could also be monitored through a GFP tag. In both cases, the mCherry signals were confined to the vacuole after growing cells until saturation in minimal selective media to maintain the plasmid (Appendix A). The signal was probably in the vacuole because the expressed peptides are not anchored in the cell in these conditions and are therefore degraded. This is consistent with the fact that Nic96 attaches to the NPC through its N-terminal domain [50], which is not present in Nic96^FAB^. Remarkably, an induction of LD content mobilization (that is, FA release) either by culture dilution, or by dilution in the presence of cerulenin, led to a solubilization of mCherry signals in the whole cytoplasm exclusively for the construct harboring Nic96^FAB^ (Appendix A). This suggests that once FAs are released from LD, the Nic96^FAB^ may be able to attach to them, which solubilizes it in the cytoplasm. As for full-length Nic96 (monitored through its GFP tag), a much fainter cytoplasmic signal (solubilization) could be seen in this experiment compared with data presented in Figure 2B. This probably reflects the out-competition of full-length by Nic96^FAB^-mCherry, which further reinforces the notion that Nic96^FAB^ may bind FAs.

Altogether then, these data point to FA-binding as a strategy to modulate Nup conformation. In particular, we suggest that free FAs, released from LD during lipolysis, could bind the core nucleoporins Nic96 and Nup157 and impart the changes needed for their release from LD. The absence of this regulation, as in *nic96* mutants lacking the protein head, would aberrantly increase Nup residence time on LD. On the contrary, the accelerated detachment provoked by cerulenin-driven FA release may trigger their unscheduled cytoplasmic solubilization.

## 4. Discussion

In this work, we have revealed the presence of nucleoporins on Lipid Droplets. Importantly, this occurs in wild-type cells and is ruled by the same determinants that define LD biogenesis. As such, the presence of nucleoporins correlates with LD formation, as occurs during late-exponential and stationary phases, upon oleate feeding, or enforced diacylglycerol formation. Reciprocally, nucleoporins become difficult to detect when the LD number is low, during early exponential growth, during LD dissolution by cerulenin, or due to LD birth difficulties caused by seipin absence or phosphatidic acid excess. Importantly, the impossibility of generating LD triggers an excess of nucleoporins at NPCs, which densely populate and even expand the nuclear membrane. Genetic interaction analyses revealed that altering LD properties this way can reconfigure the output of nuclear pore functions, such as transport and chromatin status, which therefore sensitizes cells to defects in nuclear pathways. Finally, we also provide insights into how Nup stability on LD is governed, which is notably likely through the binding of free fatty acids.

We posit that LD growth creates a landing platform capable of hosting Nups. Importantly, Nup155, the homolog of the enriched central component Nup170–Nup157, is also present in LD proteomes from fat-fed mice hepatocytes [76]. Proteins such as Pom152, whose primary structure consists of a small globular head, a single transmembrane pass, and a long luminal domain [65], will not be able to access the LD monolayer by sliding in from the contiguous Endoplasmic Reticulum. Therefore, these proteins likely accumulate at the bases of LD (Figure 9, left), as recently proposed [66], while soluble Nups could anchor to the monolayer via their amphipathic helices [67,68,69,70] (Figure 9, left) and/or through direct interaction with Pom152 moieties. In contrast, when LD contents are mobilized by lipolysis, we propose that some lipase-released FAs bind Nup157 and Nic96, helping them adopt a soluble conformation that competes with their anchorage to LD (Figure 2 and Figure 9, right). Moreover, LD dissolution forces transmembrane proteins such as Pom152 back to their regular distribution within the ER bilayer. This way, all these components may gain access to the pore, assembling into denser NPCs with altered properties (Figure 9, right) capable of differential nucleo-cytoplasmic transport patterns [77,78,79]. This phenomenon might even impact transcriptional programs and memory, DNA repair kinetics and choice, replication fork stability, and chromosome territory assemblies [3,80].

Our report constitutes the first claim of LD acting as a Nup reservoir in WT cells in a physiological manner. Up-to-date, scattered evidence throughout the bibliography points at a non-explicit link between Nup and LD in mutant contexts. For example, numerous studies report the accumulation of fluorophore-tagged Nups in the shape of cytoplasmic foci in transmembrane or inner-ring Nup-mutant backgrounds [47,48,49,50,51]. From a functional point of view, the incorrect partitioning of neutral lipids toward LD triggers defects in NPC assembly [81]. On the other hand, LD accumulate more in some Nup-deficient contexts, where their formation becomes necessary to tolerate NPC assembly stress [82]. This led to the proposal that LD supply specific lipids to conveniently remodel membranes and accommodate perturbations under incorrect NPC assembly conditions [82]. We propose the alternative, yet not exclusive, concept that LD contribute to NPC homeostasis by storing Nups. In WT cells, this would permit the pore to adopt the pertinent configuration required under each growth condition. In NPC assembly stress mutants, LD may act as a buffer for the excess of Nups that cannot be accommodated at the pore.

Along these lines, several publications document the presence of Nups in cytoplasm-related structures in other organisms. An early report concerned assembled NPCs as membranous cisternae present in germ, embryonic, and cancer cells, which are termed annulate lamellae [83]. Membrane-free cytoplasmic Nups are also described as largely soluble [84] or as condensates in stress granules during a variety of insults [85]. Very recently, an additional form of cytoplasmic Nup congregation with liquid–liquid interaction properties was found in the absence of the fragile X-related protein [86]. Yet, none of these structures are reminiscent of, and probably do not relate to, LD. There is one (partial) exception, the so-called CyPNs (cytoplasmic accumulation of promyelocytic leukemia protein (PML) and nucleoporins). CyPNs were originally described as an entity where PML would regulate the balance between the amount of cytoplasmic versus NPC-assembled Nup [87]. In subsequent works, a refined interpretation proposed that CyPNs serve as assembly bodies to prepare PML for nuclear import [88]. Thus, the presence of peripheral Nups and karyopherins in CyPN may foster PML transport through the pores [89]. CyPNs share relevant parallelisms with LD in that they host nucleoporins and karyopherins, are spherical, and, as such, gather in grape-like structures but do not aggregate. Furthermore, both are related to PML (as nuclear LD [90,91]) and move along microtubules [92]. Some major differences also exist between LD and CyPNs, for example concerning the preferential accumulation of CyPNs exclusively in the G1 phase of the cell cycle [87], or the apparent lack of scaffold Nup on CyPNs [89]. In any case, it is exciting to hypothesize that, as proposed for CyPNs, the presence of Nups on LD may help catalyze a partial pre-assembly of NPC rudiments that could help boost the resumption of functional transport once they are inserted in the nuclear membrane.

Our results also imply that conditions affecting the LD formation/dissolution equilibrium affect Nup dynamics. This may include high fat intake (Figure 4A, [9]), stresses boosting LD formation [93,94], obesity [95], aging [96,97], or degenerative diseases [98]. Furthermore, Nup dynamics is greatly affected by genetic alterations, as illustrated by *cds1* and *fld1* cells, which modify the surface and the physico-chemical properties of the LD monolayer. For example, the absence of Cds1 decreases the availability of phospholipids and thus creates monolayers with packing defects, which expose TAG to water. Nups, whose presence on LD increases in the Cds1 mutant (Figure 3), may be attracted by those exposed hydrophobic surfaces and attach to LD in a favorable reaction that suppresses packing deficiencies [6]. On the contrary, while also enlarging LD, a lack of seipin does not lead to packing defects [12], and we indeed find less Nup loading onto LD in this mutant (Figure 3). Although we attribute this decrease to the LD entanglement and difficulty of LD to emerge in the absence of seipin, our data still raise the question of whether seipin may also promote the transit and incorporation of given proteins, including Nups, onto LD [99].

Karyopherins are naturally attracted by hydrophobic FG sequences present in central ring filament Nups, and we report them as enriched on LD from both *cds1* and *fld1* mutants (Figure 6). Suggestively, apolipoproteins, the secreted LD analogues, interact with importin beta [100]. The concomitant association of karyopherin cargoes with LD (Figure 7A) very probably translates into a dramatic alteration in nucleo-cytoplasmic transport, since mutations in genes of the same nuclear pathways as the proteins sequestered onto these aberrant LD display negative interactions with *cds1* or *fld1* mutations (Figure 7B). Similarly, genetic interactions between the RSC chromatin remodeling complex and several Nup were already described [47,101], and a decreased presence of Nup170 at pores limits the access of the master transcriptional coactivator SAGA complex to its target genes, with a concomitant alteration of transcriptional profiles [102]. Importantly, we add to this view of Nup defects conditioning nuclear homeostasis by demonstrating that *cds1* and *fld1* mutations display a 7- to 10-fold increase in spontaneous nucleolar instability (Figure 7C). Thus, by involving the LD as key controllers of the NPC composition, we are placing them as masters of many genome-related transactions.

Finally, we propose FA binding as an additional mechanism contributing to the fine-tuning of Nup dynamics. Since the FAs released during LD consumption are capable of binding proteins on LD surfaces and mediate their diffusion toward (and into) the nucleus [71], we speculate that this mechanism may help complete Nup dissociation away from LD and that it might even contribute to their activation into competent forms for NPC assembly (Figure 8 and Figure 9, right).

In summary, given the profound impact that Nup availability, integrity, and stoichiometry has on cell viability and identity, the finding that their supply is ruled by LD homeostasis represents an exciting and challenging research avenue.

## Figures and Tables

**Figure 1 cells-10-00472-f001:**
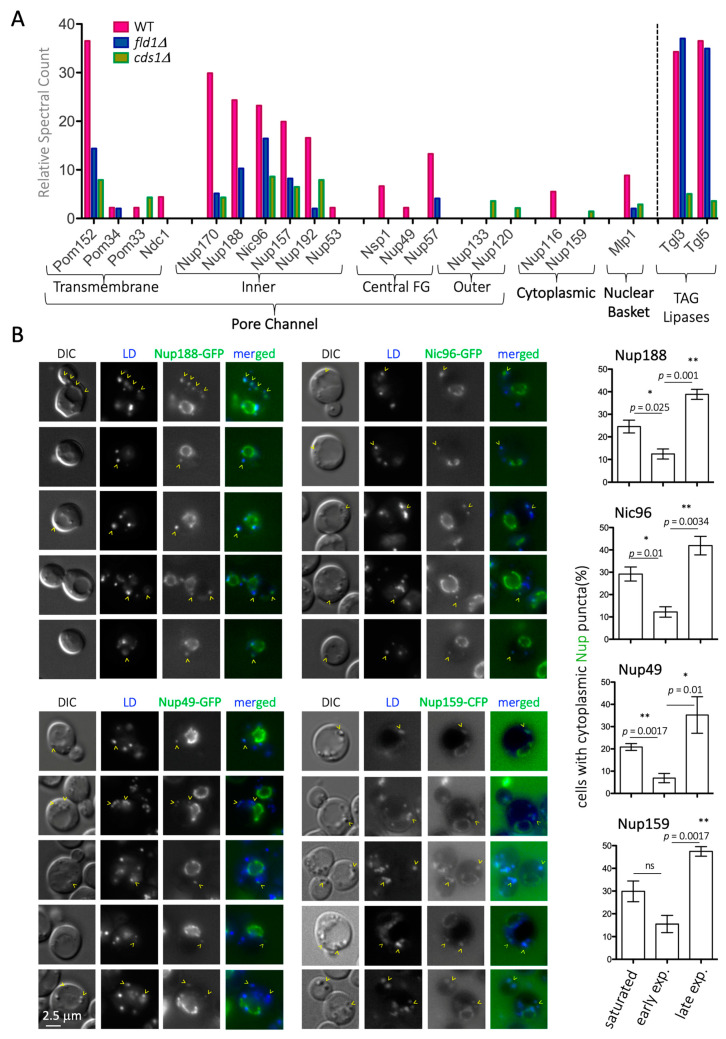
Detection of nucleoporins (Nups) on Lipid Droplets (LD). (**A**) Graphical representation of the relative presence of Nups on the LD of wild type (WT), *fld1Δ* and *cds1 Δ* cells purified from saturated *Saccharomyces cerevisiae* cultures, using raw data reported in [31]. The various Nups have been classified according to their physical positioning at the Nuclear Pore Complex (NPC). On the right, the enrichment of two LD-resident, well-characterized lipases is indicated for comparison. (**B**) **left:** Images from otherwise WT cells in which either Nic96, Nup188, Nup49, or Nup159 have been tagged with Green Fluorescent Protein (GFP) or Cyan Fluorescent Protein (CFP). The Differential Interference Contrast (DIC) images permit the inference of LD localization. AUTODOT^TM^ was used to label LD and therefore allow co-localization with Nup signals, shown in merged images. A single plane located at the mid-zone of the nucleus is shown for different cells. Yellow arrowheads point at LD in which co-localization with Nups is detected. **right:** The quantification shows the percentage of cells in the population in which LD co-localize with Nup-associated cytoplasmic signals (from one single plane). Measurements were taken in saturated, early, and late exponential cultures (see M&M for details). The graph shows the mean value, and SEM, from three independent experiments. The significance of the difference of the means from a t-test are indicated by asterisks and *p* values. ns = non-significant. At least 250 cells were considered per condition and experiment.

**Figure 2 cells-10-00472-f002:**
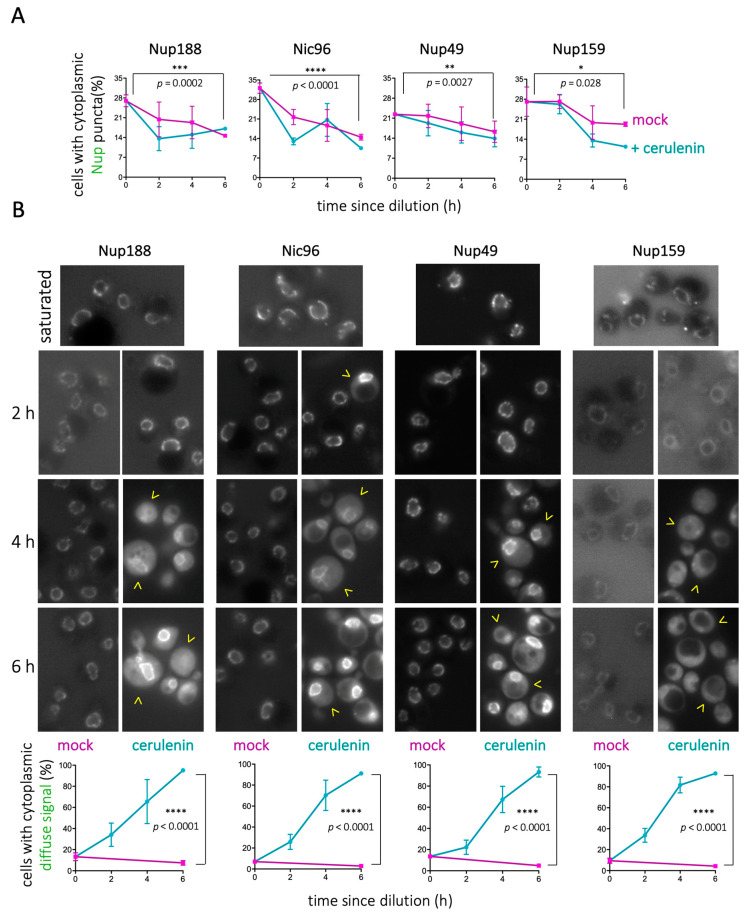
LD consumption by cerulenin dismantles cytoplasmic Nup puncta. (**A**) Overnight cultures grown in rich medium until saturation were diluted in the presence of 10 μg/mL cerulenin or in its absence (mock). Quantification showing the percentage of cells in the population displaying Nup cytoplasmic foci co-localizing with LD, for the indicated Nup, at the indicated culture timepoints. (**B**) **top:** Representative images of the timepoints described in (**A**). Yellow arrowheads point at cells in which the fluorescent signal becomes diffuse in the cytoplasm; **bottom:** Quantification showing the percentage of cells in the population in which Nup signals become diffuse in the cytoplasm. The data for each Nup are represented as an individual graph, in which timepoints correspond to the elapsed time since dilution in the presence of cerulenin or in its absence (mock). Each point is the mean value of three independent experiments. The significance of the difference of the means after applying a t-test is indicated by asterisks and P-values. Bars correspond to the SEM of those three independent experiments.

**Figure 3 cells-10-00472-f003:**
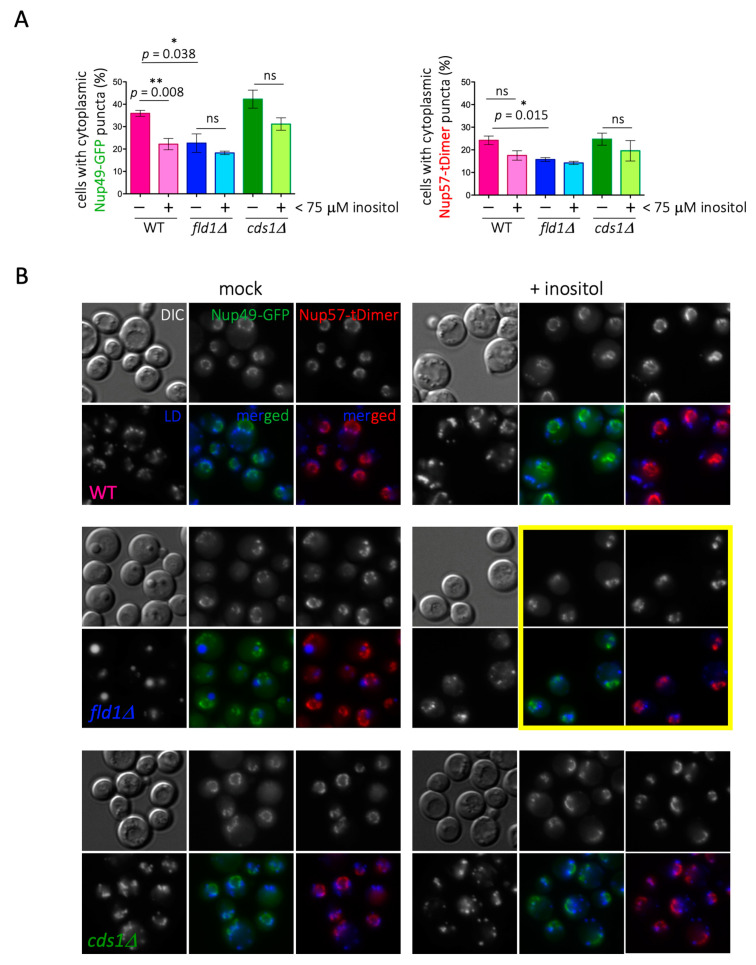
Defects in LD biogenesis alter Nup patterns. (**A**) Quantification showing the percentage of cells in the population displaying LD-co-localizing Nup49-GFP and Nup57-tDimer cytoplasmic puncta. Cells of the indicated genotypes were grown to saturation in the presence or absence of inositol. Each bar indicates the mean value of three independent experiments. Bars correspond to the SEM of those three independent experiments. Individual t-tests were applied to assess the (lack of) significance of the difference of the means, ns = not significant. (**B**) Representative images of the timepoints quantified in (**A**) for the indicated strains. The yellow frame aims at highlighting cells in which the nuclear rim became strongly deformed.

**Figure 4 cells-10-00472-f004:**
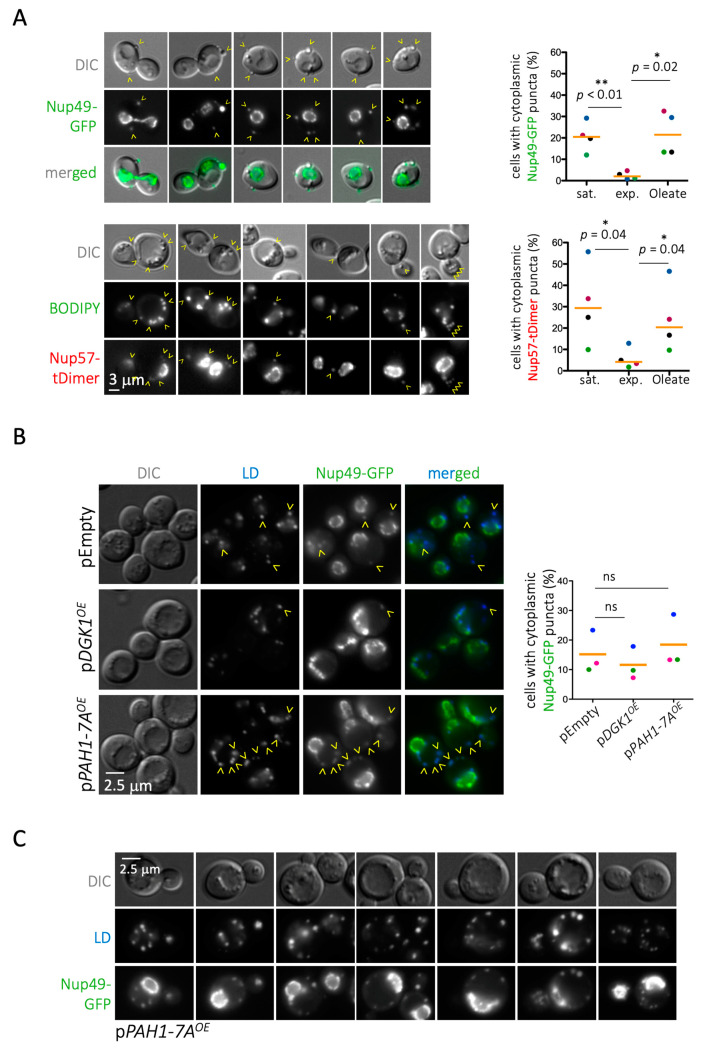
Nup patterns can be changed by dietary and genetic manipulations. (**A**) Images from WT cells fed with oleate in which Nup49 has been tagged with GFP or Nup57 tagged with tDimer. The Differential Interference Contrast (DIC) images eventually permit LD localization. BODIPY was used to label LD and thus to search for co-localization with red signals. Some Nup57–tDimer signals are overexposed at the nuclear rim to allow the visualization of cytoplasmic puncta. A single plane located at the mid-zone of the nucleus is shown for different cells. Yellow arrowheads point to LD, as inferred by DIC or marked by BODIPY, for which Nup co-localization is detected. The quantification shows the percentage of cells in the population displaying Nup-associated cytoplasmic signals (from one single plane), which in the case of the Nup57–tDimer is always coincident with BODIPY. Measurements were taken in saturated cultures and in exponential cultures grown overnight in the absence (exponential) or the presence of Oleate. The graph shows the individual points belonging to four independent experiments and the median value (orange bar). At least 250 cells were considered per condition and experiment. Dots belonging to a same experiment are shown in the same color. t-tests for paired observations were applied as indicated by bars to account for whether the different experimental set-ups led to significant changes. Results are indicated by asterisks and a *p* value. (**B**) **left:** Images from WT cells in which Nup49 has been tagged with GFP and LD dyed with AUTODOT^TM^. Cells were transformed either with an empty plasmid (control) or with a plasmid allowing the controlled overexpression of Dgk1 (increases phosphatidic acid levels) or of the constitutively active *Pah1-7A* (increases diacylglycerol levels at the expense of phosphatidic acid). Prior to imaging, the cultures, grown in glycerol overnight, were induced by adding 2% galactose for 3h. **right:** Graph showing the individual values obtained in three independent experiments and the mean value (orange bar). At least 300 cells were considered per condition and experiment. Dots belonging to a same experiment are shown in the same color. The difference of the means, after applying a t-test, was not significant (ns). (**C**) Additional images of cells overexpressing *Pah1-7A*, which increases diacylglycerol levels and prompts LD formation. LD dyed with AUTODOT^TM^ revealed that numerous Nup49 cytoplasmic signals exist in this condition and all co-localize with LD.

**Figure 5 cells-10-00472-f005:**
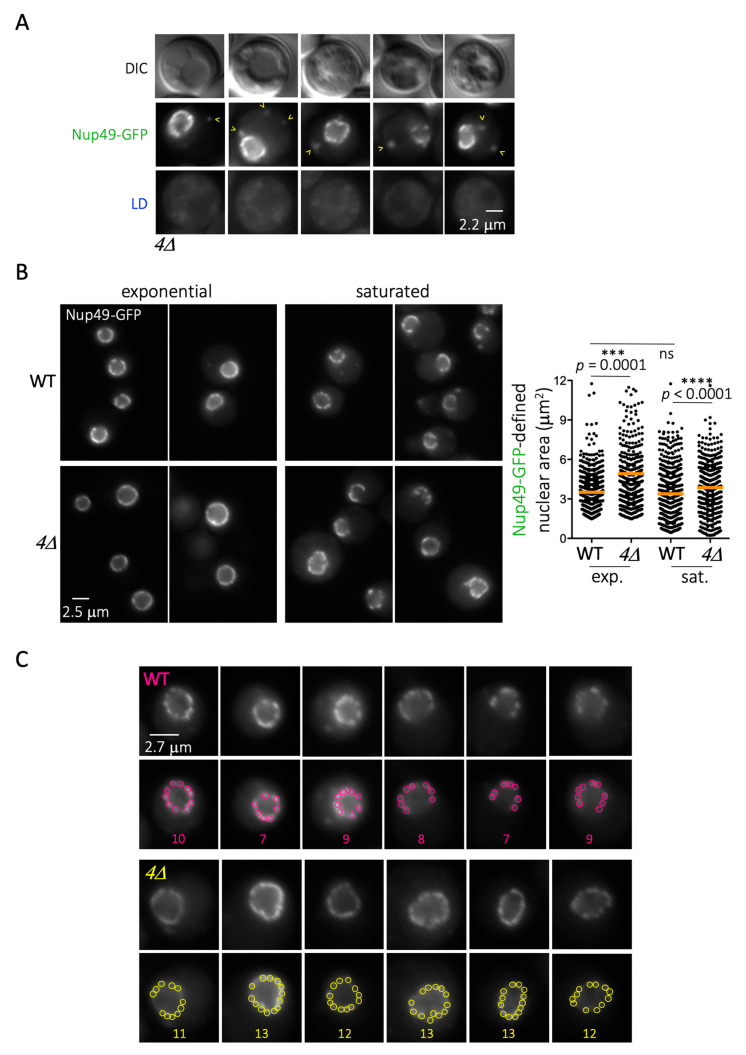
Inability to form LD triggers Nup overload at the nuclear envelope. (**A**) Fluorescence microscopy images of cells that cannot form LD (*dga**1Δ lro1Δ are1Δ are2Δ*, simplified as *4Δ*) in which Nup49 has been tagged with GFP. (**B**) **left:** Fluorescence microscopy images of WT and *4Δ* cells in which Nup49 has been tagged with GFP, grown either to exponential phase or to saturation. **right:** Plot of the sizes of individual nuclear areas as defined by the Nup49-GFP rim for each strain for the indicated growth set-ups. Two independent experiments are included per condition in the plot. At least 300 cells were analyzed per condition and experiment. The orange bar indicates the mean of the population. The *p* values refer to the statistical significance of the difference of the means by t-test. (**C**) Images of individual nuclear rims of both WT and *4Δ* cells from saturated cultures at the central plane only. For each strain, the upper row corresponds to the unprocessed image, while the lower row shows that same image, to which colored circles have been superimposed. Each circle comprises an NPC cluster, as identified following the directives presented in [34,35,59]. The numbers indicate the number of individual clusters identified for each nuclear rim.

**Figure 6 cells-10-00472-f006:**
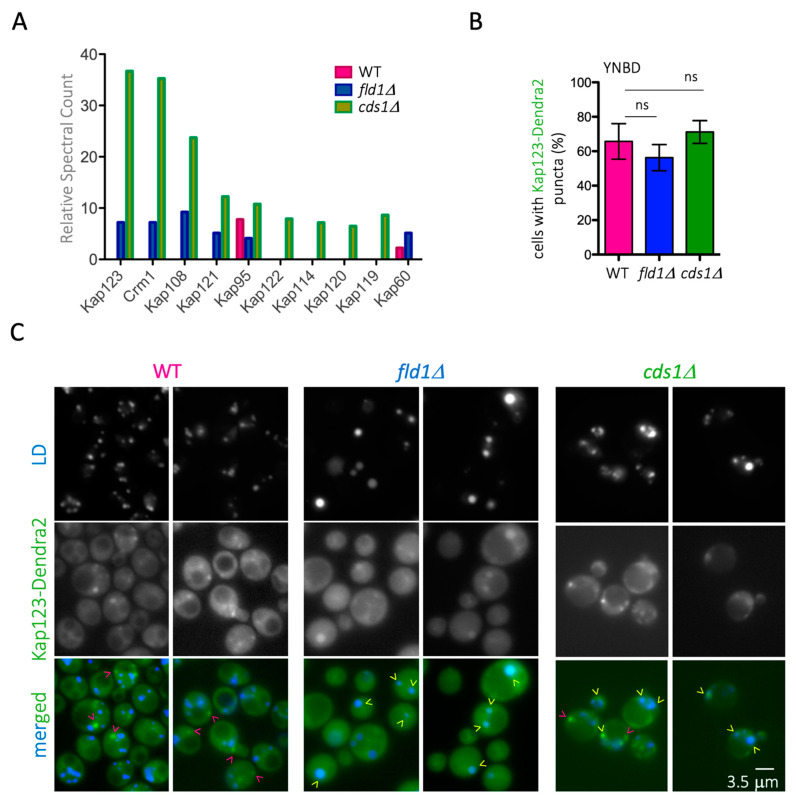
Altered distribution of karyopherins onto mutant LD. (**A**) Graphical representation of the relative presence of karyopherins on the LD of WT, *fld1Δ*, and *cds1Δ* cells purified from saturated *Saccharomyces cerevisiae* cultures (raw data reported in [31]). (**B**) Quantification of WT, *fld1Δ*, and *cds1Δ* cells with Kap123–Dendra2 cytoplasmic puncta upon growth in minimal, defined medium until saturation. Each bar indicates the mean value of three independent experiments. Bars correspond to the SEM of those three independent experiments. The difference of the means after applying a t-test revealed that it was not significant (ns). (**C**) Fluorescence microscopy images from data presented in (**B**) of WT and mutant cells in which Kap123 has been tagged with Dendra2 (2 examples per genotype). AUTODOT^TM^ was used to dye LD. Merged images are shown, to assess eventual co-localization. Yellow arrowheads indicate Kap123–Dendra2 signals that co-localize with LD, while pink arrowheads indicate those that do not co-localize.

**Figure 7 cells-10-00472-f007:**
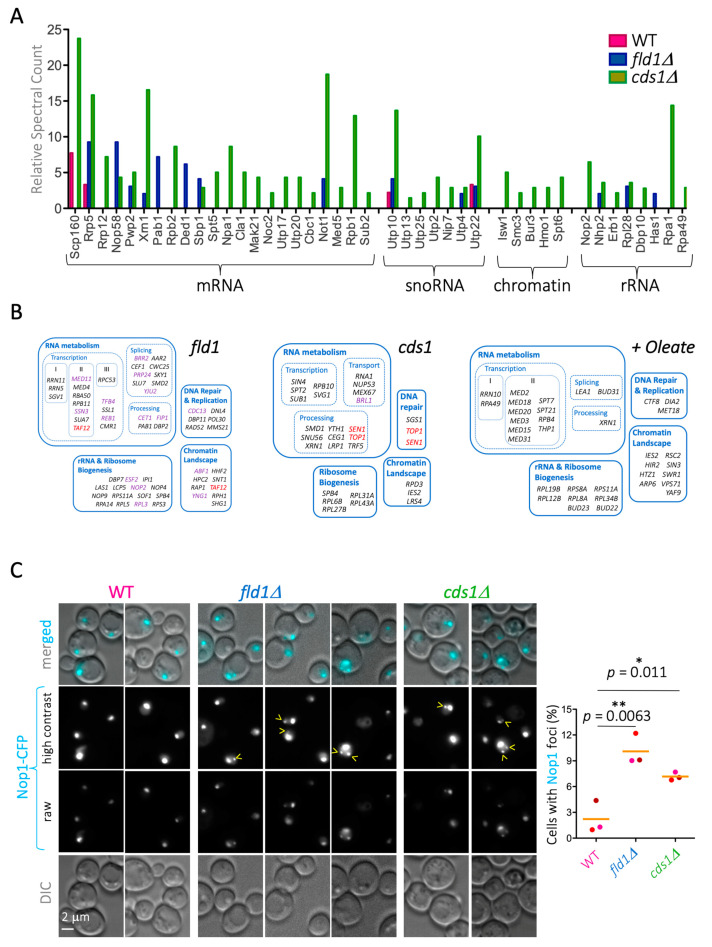
LD alterations predispose to genome instability. (**A**) Graphical representation of the relative presence of karyopherin cargoes found on the LD of WT, *fld1Δ*, and *cds1Δ* cells purified from saturated *Saccharomyces cerevisiae* cultures (original data reported in [31]). The hits are grouped by biological role. (**B**) The 257 genetic interactors of the *fld1* mutation (left) and the 232 interactors of the *cds1* mutation (center) were retrieved from the *Saccharomyces* Genome Database, while the 134 oleate-sensitive interactors (right) were retrieved from [32]. The nuclear hits were selected, and further functional classification was performed with DAVID [33], and these are shown as individual sets. Genes in violet indicate positive genetic interactions, while the rest are negative ones. Genes in red are included twice because the protein can be ascribed to two functional categories. (**C**) **left:** Images of WT, *fld1Δ*, and *cds1Δ* cells that were transformed with a plasmid expressing the nucleolar marker Nop1 tagged with CFP. Asynchronous cultures were used for live fluorescence microscopy. Two or three examples are shown for each strain. Nucleoli are visible in the raw and high-contrast images. Nop1 foci, indicative of fragmented nucleoli, are indicated by yellow arrowheads. **right:** Quantification of the percentage of cells in the population with Nop1 foci. Dots represent three independent experiments. Similarly colored dots belong to the same experiment. The orange bar is the mean of those three experiments. At least 300 cells were counted per condition and experiment. The *p* values correspond to paired t-tests assessing the significance of the differences of the means.

**Figure 8 cells-10-00472-f008:**
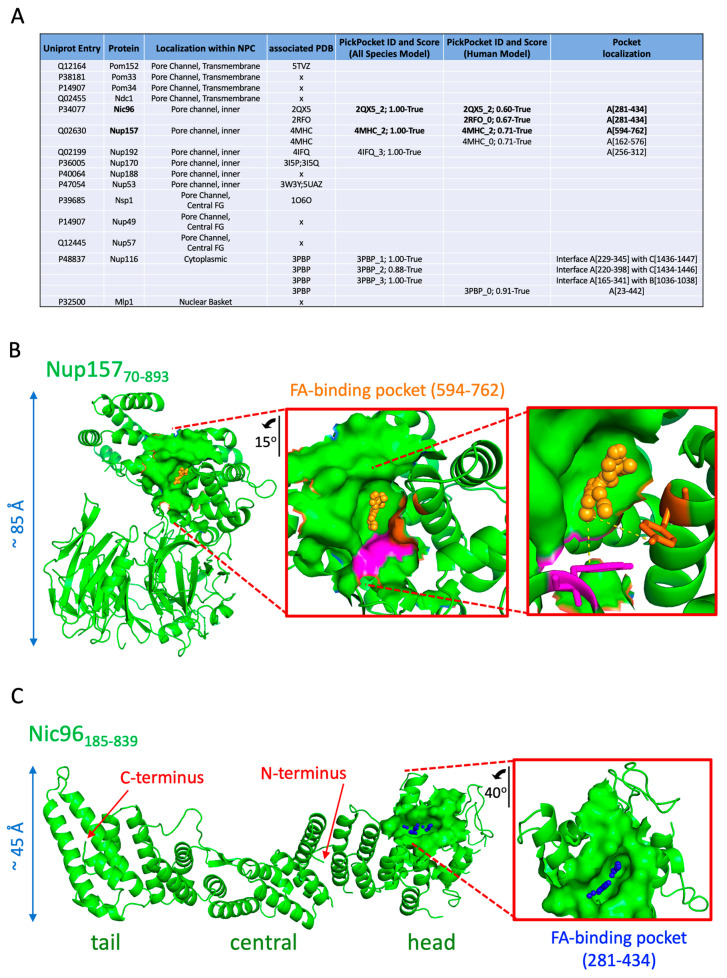
Prediction of fatty acid (FA)-binding pockets in Nups. (**A**) Table compiling the prediction information for all the Nups present on WT LD. Indicated are the presence (or absence, x) of an available PDB ID to be used for the prediction. Only positive results (where “False” is assigned to scores below 0.5, while “True” spans up to the perfect match of 1, see M&M) are indicated upon running PickPocket, which is either trained with an All Species Model or a Human-Only Model. The protein chain and the aminoacids stretch harboring the predicted pocket are also shown. (**B**) Structure of Nup157 (aminoacids 70-893; PDB 4HMC), in which the position and orientation of the predicted FA-binding pocket is highlighted as a filled space. An inset of the coordination of the FA with Tyr594 (magenta) and Tyr646 (orange) is shown. (**C**) Structure of Nic96 (amino acids 185-839; PDB 2QX5), in which the position and orientation of the predicted FA-binding pocket is highlighted as a filled space. The different domains of the protein have been named for clarity.

**Figure 9 cells-10-00472-f009:**
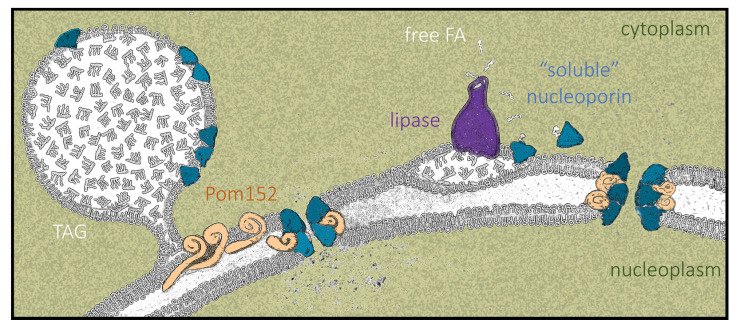
Model of how LD status impacts Nup configurations at the nuclear pores. **left:** TriAcylGlycerols (TAG)-filled LD create a landing platform for nucleoporins, which can attach by their own features (i.e., amphipathic helices), by interaction with other Nups, or by congregation in the perimeter of LD, e.g., in the case of Pom152. This may titrate nucleoporins away from the nuclear pore, defining a particular NPC stoichiometry that matches a specific transport and chromatin landscape profile. **right:** In the event of LD consumption, lipase activity releases individual fatty acids. We propose that these FAs may bind core Nups of the inner pore channel, such as Nic96 and Nup157, promoting a conformational change that permits their dissociation from LD and migration to the pore. Simultaneously, the decreased surface of the LD upon shrinkage will naturally promote the eviction of other Nups, as ruled by protein crowding [53], as well as the diffusion back to the ER of transmembrane proteins congregating in the vicinity of the LD [66]. The availability of these components at the NPC will specify a different profile of transport and alternative chromatin features.

**Table 1 cells-10-00472-t001:** Strains used in this study.

Simplified Genotype	Full Genotype	Source
WT	*MAT*a, *ade2, his3, can1, leu2, trp1, ura3*, GAL+, psi+, *RAD5+*	PP870, Philippe Pasero
Nic96-GFP	*MAT*a, *ade2, his3, can1, leu2, trp1, ura3,* Nic96-GFP::*kanMX6*	Y4057, Ed Hurt
Nup188-GFP	*MAT*a, *ade2, his3, can1, leu2, trp1, ura3,* Nup188-GFP::*kanMX6*	Y4070, Ed Hurt
Nup49-GFP	*MAT*a, *ade2, his3, leu2, trp1, ura3*, *ARS604*::LacOp-*TRP1*, Nup49-GFP	PP3188, Armelle Lengronne
Nup159-CFP	*MAT*a, *ade2, his3, can1, leu2, trp1, ura3, kanMX6,* Nup159-CFP::*natNT2*	Y4405, Ed Hurt
Nup57-tDimer	*MAT*a, *ade2, his3, leu2, trp1, ura3*, Nup57-tDIMER::*LEU2*	PP3773, Armelle Lengronne
*fld1* *Δ*	*MAT*a, *ade2, his3, can1, leu2, trp1, ura3*, *fld1**Δ**hphMX4*	This study
*cds1* *Δ*	*MAT*a, *ade2, his3, can1, leu2, trp1, ura3*, *cds1**Δ**kanMX6*	This study
Kap123–Dendra2	*MAT*a, *ura3**Δ**, leu2**Δ**, his3**Δ**, met15**Δ**, bar1**Δ**HIS5,* Kap123*–*Dendra2*::natNT2*	Marta Radman-Livaja
*4* *Δ*	*MAT* a *, ade2, ura3, can1, dga1* *Δ* *kanMX6 lro1* *Δ* *TRP1 are1* *Δ* *HIS3 are2* *Δ* *LEU2*	H1246, Zvulum Elazar

**Table 2 cells-10-00472-t002:** Plasmids used in this study.

Simplified Name	Detailed Information	Source
pEmpty	YEplac181	Symeon Siniossoglou
p*DGK1*	YEplac181-*GAL1/10p-DGK1*	Symeon Siniossoglou
p*PAH1-7A*	YEplac181-*GAL1/10p-PAH1-7A*	Symeon Siniossoglou
p*SEC63*	pJK59 (*CEN-URA3-SEC63p-SEC63-GFP*(S65T,V163A))	Sebastian Schuck
p*NOP1-CFP*	p*NOP1-CFP::LEU2*	Danesh Moazed
p*NIC96^FAB^-mCherry*	pRS316-*CYC1p-NIC96^FAB^-mCherry-NUP1t*	This study
p*mCherry*	pRS316-*CYC1p-mCherry-NUP1t*	This study

## Data Availability

All the information concerning this work is included within this manuscript or attached as Appendix A.

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
