# Peer review of "Lipid Droplets Are a Physiological Nucleoporin Reservoir"

_cells, 2021, doi:10.3390/cells10020472_

Round 1
Reviewer 1 Report
In this study, Kumanski et al. set out to investigate the link between lipid droplet (LD) formation and nuclear pore complex assembly. They propose an interesting model where LDs recruit a subset of NPC proteins thereby restricting their localization and affecting their availability to form NPC. LDs have been recently proposed to function in a similar way for other proteins and transcription factors. As the authors pointed out, NPC proteins have been previously reported to reside on LDs, but the functional implications of this were poorly understood.
The study is interesting, it is well-put-together and easy to read. The data is clear, and the proposed model is logical although in some instances there are overstatements of conclusions especially regarding the fatty acid-binding data. A weakness of this work is that the authors rely too much on published proteomics results and public databases such as SGD to prove their hypothesis without performing additional experiments.
The study while interesting and provocative as it shines the light on an area that is less understood in LD research, it is also largely correlative and lacks further experimental evidence to substantiate some of the claims. More experiments such as imaging and analysis of mutants could greatly improve the quality and impact of this study. Please see the comments below:
Major points:
1- The role of lipolysis in regulating Nup localization is not sufficiently explored. The authors claim that the localization of Nup proteins to LDs might be regulated by the metabolic status of the cell. Does starvation or addition of drugs that stimulate LD lipolysis such as cerulenin affect Nup localization to LDs? Feeding oleate promotes LD formation and also promotes the association of numerous hydrophobic proteins to LDs and thus not the best way to support this claim.
- Figure 1B lacks LD stain. A panel for LD staining (such as MDH or AutoDot) needs to be added in order to conclude on the co-localization of LDs.
- Can the authors comment on why they chose to image the central FG Nups and not the inner channel Nups? This is especially important because, at the end of the study, they focus on the inner Nup157 and Nic96.
- Images of oleate fed yeast corresponding to graphs in Figure 1B, C need to be added with tagged NUP and LD stain
2- The roles LDs/lipid storage and phospholipid/membrane homeostasis in regulating Nup localization are not clear.
First, the relationship between phospholipid synthesis and Nup localization to LDs is not clear.
It is also unclear how Fld1 deletion causes a reduction in Nup association with LDs. Can the authors comment on why that is? Are ER-LD connections needed?
Imaging to confirm the localization of Nups in the Cds1 and Fld1 mutants is needed. Also, a proper definition of Cds1 and Fld1 in the text is needed.
The authors claim that PA accumulation in the Fld1 and Cds1 deletion affects LD-ER connections and thus somehow affect Nup localization to LDs. PA accumulation also affects nER morphology. Do they see changes in nER morphology? Does Pah1 deletion (which converts PA->DAG) have the same effect? Is this reversed by Dgk1 overexpression? Dgk1 converts DAG->PA.
The authors show that Seipin deletion causes supersized LD and that is one of the factors that caused Nup recruitment. Adding inositol can reverse supersized LD morphology. Does that affect Nup localization?
3- Fatty acid-binding and Nup release are not sufficiently explored. Predicted FA binding sites were found in the inner channel Nups but not the ones they imaged in Figure 1 (Nup49 and 57)?
Does mutation of FA-binding residues affect localization to LDs?
Does Nic96 head domain by itself localize to LDs?
Imaging to show that mutants lacking head domain don’t localize to LDs or aren't properly released from LDs will help clarify this section.
Minor points:
- Figure 1 B, C it seems only LDs near new bud sites have the Nup signal. Is there a relationship between Nup localization and budding? Or nuclear inheritance?
- Citing the Mol Cell paper on MLX-Family Transcription Factors by Mejhert et al would be relevant and appropriate.
- Line 286 “we ran” instead of “we run”
- Lower caps Fatty Acids (fatty acids)
- Line 348 instead of “Authors” use “Previous work” or a similar phrase.
- Figure 1B,C graphs y-axis: Spectral counts instead of Spectrum counts.
- Figure 1B,C graphs y-axis: Be more specific about which Nup foci that were quantified in the corresponding graphs
Reviewer 2 Report
Comments:
- The material and methods section need to be more descriptive.
- It is not clear what the authors mean by spectral counts.
- There seems to be curation of previously published data and from information available on SGD. Authors must validate this information present it in the form of data.
- Figure 1B, authors must use a dye to stain LDs.
- Model for fatty acid binding pockets is interesting. Authors must validate and elaborate this.
Reviewer 3 Report
Kumanski et al. propose nucleoporins to relocalize to lipid droplets, which function as a physiological depot storage site. By reanalyzing a previously published dataset, they find various proteins of the nuclear pore complex to be identified in the LD proteome, which is lost in the absence of Cds1p or Fld1p (yeast Seipin). The presence of Nup49 and Nup57 on LDs is verified by fluorescence microscopy. By further scrutinizing the LD proteome, they identify nucleoporins to localize to LDs specifically to supersized LDs (in fld1Δ or cds1Δ cells). Finally, using an in silico approach, a fatty acid binding pocket is predicted in different nucleoporins, which the authors propose to mediate LD binging.
The authors conduct a thorough re-analysis of published data, in which they find indications for NPC protein localization to LDs. Based on the proteome data, they correlate the presence of nucleoporins on LDs to LD size. As the dynamic LD proteome still remains elusive, these observations are of interest to the field. However, there are several points that should be addressed to see this manuscript fit for publication. In general, LD localization is only validated in vivo for two proteins, and not all correlations proposed by the authors are supported by experimental evidence. The identified fatty acid binding pocket is not validated experimentally, and the authors do not show indications for a physiological role of LD localization of nucleoporins.
Specific issues to improve the manuscript:
- The fluorescence microscopy imaging in Figure 1 should be improved. To support LD localization of Nup49, a lipid droplet marker (e.g. the blue fluorescent neutral lipid dye MDH, or Erg6-RFP) should be included in Figure 1B. In Figure 1C, the signal of Nup57-tDimer seems overexposed, making it difficult to assess colocalization in the merge BODIPY/tDimer panel.
- Loss of Nup-protein localization to LDs in fld1Δ or cds1Δ should be evidenced by microscopy imaging.
- The localization of nucleoporins to supersized LDs in the absence of Fld1p or Cds1p should be shown experimentally.
- The PickPocket software used for the prediction of fatty acid binding pockets (as shown in Figure 4) is not published yet, and the authors refer to a pre-print. In that light, the authors should describe the algoritm in more detail, and should show controls with regards to the use of this software.
- The proposed role of the fatty acid binding pockets in LD localization of these proteins should ideally be addressed experimentally, in example by mutating residues that bind the fatty acids in this pocket and investigating the effect on LD localization. To substantiate the proposed physiological role of LD localization of these proteins, the following points should be addressed or discussed:
-
- What is the fate of these proteins in the absence of LDs (in example in a dga1Δlro1Δare1Δare2Δ background strain, or upon stimulation of LD turnover by cerulenin treatment)?
- What is the function of LD localization of these proteins? How does this contribute to nuclear homeostasis?
Round 2
Reviewer 3 Report
The revised manuscript by Kumanski et al. is substantially improved with new data in support of the conclusions. The authors propose that a small fraction of nucleoporins co-localize with lipid droplets, which may serve as storage site for those proteins. The original version was heavily based on reanalysis of a previously published dataset, which lacked stronger physiologically relevant findings. Also, using an in silico approach, a fatty acid binding pocket was predicted in different nucleoporins, which the authors propose to mediate LD binding – though no in vivo or in vitro data was shown to substantiate this claim.
In the revised manuscript the authors have conducted additional experiments and added key controls resulting in great improvements.
Overall, the findings are of importance to the field and the manuscript is suitable for publication in Cells.
However, there are a few minor issues that when corrected would improve the manuscript.
- Statistics – there is a lack of cohesion with the statistical analysis. Some figures show statistical analysis and some do not (e.g. Figure 1B).
Also, It is strange that SEM is used for 3 repeats, rather than SD. However, if the authors decide to do so, they just need to make sure this is consistent throughout the article.
- English – The choice of words and phrases in the manuscript quite strange, for example the first paragraph of section 3.3 and the use of the word “emancipate” in the last line of section 3.3.
I would suggest to let a native English speaker go over the text to polish it up and allow for an easier read.
- The findings in figure 9 are puzzling. I appreciate the attempt to validate the FAB binding site in vivo, though the data presented is not conclusive. The authors rely on a diffused nic96 FAB mCherry signal as their readout, using mCherry and nic96 WT as controls.
In general, the data is of good quality. However, it is puzzling that Nic96 WT-GFP doesn’t follow the same trajectory as the nic96 FAB. Are there other known regions keeping the protein in the nuclear envelope? If FAB is important then why does the WT protein not change its localization upon addition of cerulenin? Is it because it is in the NE or LDs? The authors site reference 50 in their manuscript as showing that Nic96 lacking FAB is in punctate form in the cytoplasm but this is not discussed properly in light of the findings in this paper and how this fits with the current findings is unclear. Though I do not think more data/controls are needed at this point, these points raised need to be discussed properly.
